# Human–Robot Collaborations in Smart Manufacturing Environments: Review and Outlook [note 1]

**DOI:** 10.3390/s23125663

**Published:** 2023-06-17

**Authors:** Uqba Othman, Erfu Yang

**Affiliations:** Department of Design, Manufacturing and Engineering Management, University of Strathclyde, Glasgow G1 1XJ, UK

**Keywords:** human–robot collaboration (HRC), industry 4.0, smart manufacturing, literature, artificial intelligence (AI), digital twin (DT), collaborative robot (Cobot), augmented reality (AR)

## Abstract

The successful implementation of Human–Robot Collaboration (HRC) has become a prominent feature of smart manufacturing environments. Key industrial requirements, such as flexibility, efficiency, collaboration, consistency, and sustainability, present pressing HRC needs in the manufacturing sector. This paper provides a systemic review and an in-depth discussion of the key technologies currently being employed in smart manufacturing with HRC systems. The work presented here focuses on the design of HRC systems, with particular attention given to the various levels of Human–Robot Interaction (HRI) observed in the industry. The paper also examines the key technologies being implemented in smart manufacturing, including Artificial Intelligence (AI), Collaborative Robots (Cobots), Augmented Reality (AR), and Digital Twin (DT), and discusses their applications in HRC systems. The benefits and practical instances of deploying these technologies are showcased, emphasizing the substantial prospects for growth and improvement in sectors such as automotive and food. However, the paper also addresses the limitations of HRC utilization and implementation and provides some insights into how the design of these systems should be approached in future work and research. Overall, this paper provides new insights into the current state of HRC in smart manufacturing and serves as a useful resource for those interested in the ongoing development of HRC systems in the industry.

## 1. Introduction

Human–robot collaboration (HRC) is an essential area of focus in the smart manufacturing industry. The automotive and food sectors are among the industries with the most significant need for HRC systems. Traditionally, robots have been utilized in manufacturing to perform repetitive and simple tasks. However, with advancements in technology, researchers are now exploring ways to integrate human expertise, decision-making, and critical thinking with the strength, repeatability, and accuracy of robots to perform complex tasks.

With the increasing adoption of automation technologies in manufacturing, robots are expected to work alongside humans to achieve higher levels of productivity, quality, and flexibility. HRC has emerged as a promising approach to achieving these goals. By working collaboratively, humans and robots can combine their unique strengths and abilities to optimize manufacturing operations. Therefore, the utilization of HRC in manufacturing is changing the traditional approach of utilizing robots, and researchers are increasingly focusing on exploring new ways to exploit human–robot collaboration to enhance manufacturing efficiency and quality [1].

Recent studies have highlighted the benefits of human–robot collaboration in smart manufacturing. One of the key benefits is the improved productivity. By leveraging the strengths of robots and humans, manufacturers can achieve higher production rates and faster cycle times. [2]. Recently, the demand for customized products has increased and, eventually, customized manufacturing also increased to meet this demand. Therefore, manufacturers are modifying their work environments to be more intelligent and reliable. This has resulted in the creation of human–robot collaboration systems in smart manufacturing, where robots and humans operate together harmoniously to achieve better productivity and faster cycle times, while preserving a secure and efficient workplace. By utilizing their respective strengths, robots can take on tasks that are monotonous or hazardous, while humans can handle more intricate and innovative tasks. These systems’ abilities have been further enhanced by sophisticated technologies such as machine learning and artificial intelligence, allowing robots to learn from humans and adjust to shifting circumstances. The inclusion of human–robot collaboration systems has revolutionized the industry, giving manufacturers a way to boost productivity, efficiency, and workplace safety [3].

The following paper provides an in-depth review of the current state of research in human–robot collaboration in smart manufacturing. In particular, it attempts to define and classify human–robot interactions in terms of work tasks, direct contact, and simultaneous and sequential processes, as well as human–robot collaboration in terms of collaboration levels, work roles, safety control modes and communication interfaces. Then, it investigates the recent technological advances in implementing HRC systems and illustrates some industrial applications. Key challenges of HRC are highlighted and used to identify future research directions.

### Research Methodology

This research article presents an extended version of a conference paper [4], aiming to provide a comprehensive analysis of human–robot collaboration in smart manufacturing. The methodology employed in this study involved a literature review and an expansion of the previous conference paper’s content. The following steps outlined in Table 1 were taken to gather the necessary information in order to answer the research questions.

## 2. Human–Robot Interaction Classification

Industrial robots have become an essential component in the race among companies to improve production efficiency. The International Federation of Robotics projected a 13% growth rate for the robot production industry worldwide in 2019 [5]. Furthermore, numerous corporations are placing their attention on incorporating distinctive capabilities into the robotic systems that they are considering. These special features may include the need for robots that are more user-friendly, flexible, and secure. As a result, there is a growing demand for robots that can work alongside humans without posing a threat to their safety, while being able to adapt to a wide range of tasks and environments [6]. As collaborative robots, also known as “cobots,” continue to gain popularity, businesses of all sizes are finding it increasingly feasible to incorporate robotic systems into their production processes. This, in turn, can lead to improved efficiency and flexibility within industrial environments. The integration of human–robot interaction phases plays a crucial role in this process, as it allows for more seamless collaboration between the robot and the human operator. For instance, cobots can be programmed to perform repetitive or physically demanding tasks, freeing up human operators to focus on more complex and creative aspects of the production process. Ultimately, the successful integration of cobots can result in a safer, more productive, and more adaptable industrial ecosystem [7]. The nature of human–robot interaction is heavily influenced by various factors, including the specific task that needs to be performed, the shared workspace, the degree of direct contact between the human and the robot, as well as the sequencing and timing of the different processes involved. Consequently, interactions between humans and robots can be broadly categorized into four primary types [8,9]:Coexistence interaction: This interaction refers to a scenario where a human operator and a robot are working on different tasks in different workspaces without the need for physical barriers [10,11]. For example, robots may be responsible for heavy lifting and assembly while human operators oversee quality control and oversight. The limited connection between the human and the robot in this type of interaction allows for greater flexibility and efficiency in the production process [10].Synchronization interaction: This type of interaction between the human and the robot involves a scenario where a human operator and a robot share the same workspace, but work at different times in a sequential manner. Both the human and the robot are responsible for performing specific tasks, and they communicate with each other by providing instructions and feedback. This type of interaction requires a high degree of coordination and synchronization between the human and the robot. In this scenario, the human operator and the robot are looking at the same target and working in sequential order to achieve the desired outcome. So, the human operator might be assigned for loading a machine with raw materials, while the robot is performing the actual manufacturing process. The human operator and the robot would need to work together in sync to ensure that the machine is loaded correctly and that the manufacturing process is carried out without interruption [8].Cooperation interaction: Cooperation relation refers to where a human operator and a robot work at the same time towards a shared objective but have separate interests. They both have access to the same technological resources to obtain information about the work task, but there is no direct connection between them. Even though their workspaces may overlap, the human operator and the robot do not interfere with each other’s work. The focus is on achieving a common goal while pursuing individual interests. For instance, in a warehouse, a human operator might be responsible for managing inventory and order fulfilment, while a robot is responsible for material handling and transportation. Both the human operator and the robot can access the same information on inventory and order status but work independently to achieve their respective objectives. This type of interaction promotes efficient resource allocation and coordination between the human and the robot.Collaboration interaction: Collaboration interaction involves a scenario where a human operator and a robot work in synergy towards a common goal in the same workspace at the same time. This type of interaction is more advanced than cooperation interaction and requires a high level of coordination and communication between the human operator and the robot [12]. In collaboration interaction, the human operator and the robot work closely together, and direct contact between them is possible and under their control through advanced sensing technologies, as the action of one has immediate effect on the other. Considering that the human operator and a robot might work together to assemble a complex component, the human operator might be responsible for the more delicate aspects of the task, such as placing, aligning components and making decisions [13], while the robot might be responsible for heavy lifting and precise positioning. The success of collaboration interaction depends on the ability of the human operator and the robot to work together seamlessly and efficiently. The connection between the human and the operator can consist of both physical and contactless connection, where different information may be required to meet the requirements of the connection type in this type of interaction. Physical connections can be carried out between the human and the robot by measuring forces and torques, then the robot will be able to understand the human intentions and take action accordingly [14]. At the same time, a contactless connection between the human and the robot is implemented through appropriate communication techniques to ease the working relationship between the human and the robot. Direct (speech, gestures, force) and indirect (intention recognition, eye blinking) communications between the human and the robot, can be detected and analyzed using advanced sensing technologies such as machine vision and haptic feedback, so the robot will be able to understand the human’s intentions and respond regarding the task needs [15].

Table 2 summarizes human–robot interaction features considering the shared contents of work tasks, direct contact, and simultaneous and sequential processes [16].

## 3. Definition and Classification of Human–Robot Collaboration

In HRC systems, the focus is on combining the strengths and abilities of both the human operator and the robot to achieve a common work target. The main aim of HRC is to create a system where the human operator can leverage his/her experience, judgment, and decision-making skills, while the robot can contribute its speed and precision. At the same time, it is mandatory to ensure that the human operator and the robot are physically separated during work to avoid potential hazards or accidents [17]. Collaboration interaction in HRC systems is a game-changer for industrial processes. By allowing human operators and robots to work together in direct connection, collaborative working areas can facilitate better communication, information exchange, and joint targeting of tasks. This approach allows for better productivity and work efficiency. The growing interest in developing collaborative working areas is a testament to the potential benefits of HRC systems.

As companies increasingly invest in these systems, they are better able to leverage the strengths of both human operators and robots. The ability of human operators to share and exchange information with robots allows for a more flexible and adaptable workflow, as the system can learn from the experiences of human operators and adapt to changing conditions. This, in turn, leads to better productivity and more efficient production processes. Furthermore, collaborative working areas enable a more seamless integration of human operators and robots, as the system agents can work together in the same workspace without causing interference. This approach not only improves productivity but also enhances safety, as robots can support human operators in risky or physically demanding tasks [18]. This indicates the flexibility of creating a direct connection between the human operator and the robot [19].

HRC systems are designed to work intelligently as the production objectives are delivered feasibly. Technological enhancements proposed in smart manufacturing areas allow the human operator work competently [20]. The integration of collaborative robots in HRC systems provides an efficient and flexible production process, enabling human operators to make crucial decisions and impact the entire production process. Collaborative robots possess intuitive interfaces and sensory systems that allow them to assist human operators with repetitive and hazardous tasks, thereby enhancing productivity and reducing the risk of workplace injuries. This cooperation between human operators and robots leads to an optimized production process, where both parties can leverage their strengths and abilities to improve productivity and efficiency in the manufacturing industry [21]. The Fourth Industrial Revolution has brought about significant improvements to conventional programming approaches. As a result, even individuals who lack technical expertise can effectively operate and communicate with robots. Utilizing simple gestures, voice commands, and even eye movements, workers can interact with robots during tasks without the need for traditional tools. This shift towards more intuitive and natural means of communication has transformed the nature of work from reactive to proactive. The enhanced capabilities of robots have enabled them to work alongside humans collaboratively and efficiently, driving progress and innovation in various industries. [3]. Human–robot collaboration is determined by the agent’s effort dynamics, the nature of work concerns, human operator satisfaction, and the ease of critical information transferring between operators and cobots [22]. Accordingly, the HRC system can be classified into four main aspects that are outlined in Figure 1.

### 3.1. Collaboration Levels

The HRC system is initiated on the principle of teamwork and cooperation between humans and robots. To facilitate effective collaboration in shared environments, researchers have been working to establish standardized approaches. In [22,23], the levels of collaboration in the HRC system have been categorized as follows:Independent: When humans and robots collaborate at an independent level, they concentrate on different tasks individually. The robot carries out its allocated duties without human intervention, while the human operator works on tasks separately. Although there is no direct interaction or cooperation between them, the robot’s presence can improve productivity and efficiency in the workflow. This level of collaboration is ideal for tasks that do not need close coordination or interaction between humans and robots.Sequential: The HRC system involves both humans and robots sequentially collaborating on tasks. The robot performs its designated task first, followed by the human operator, who then completes the human part of the task. Coordination and communication between the human operator and the robot are essential to ensure a smooth and uninterrupted workflow. Sequential collaboration is ideal for tasks that require a combination of human decision-making and robot precision. For instance, in a manufacturing process, the robot may perform a precision task like welding, while the human operator inspects or performs a quality check after welding. This collaboration level can enhance productivity and efficiency by reducing task completion time and minimizing errors.Simultaneous: When humans and robots collaborate at a simultaneous level, they work on the same task but with different processes. The human operator and the robot perform their respective tasks simultaneously, without direct coordination or cooperation. For instance, in a manufacturing process, the robot may perform a task such as cutting or drilling, while the human operator performs a task such as loading or unloading materials. Although they work on the same task, they do not need to communicate or coordinate their actions with each other. This level of collaboration is best suited for tasks that don’t require close coordination or communication between humans and robots. By enabling both the robot and the human operator to work on the same task simultaneously, this level of collaboration can increase productivity and efficiency, thereby reducing the overall time required to complete the task.Supportive: In the HRC system’s supportive level of collaboration, the operator, and the robot work together in synchronized harmony to perform a common process on the same workpiece. This kind of collaboration requires a high degree of coordination and communication between the human operator and the robot, since both parties are simultaneously engaged in the same task. For instance, in manufacturing, while the human operator works on tasks such as welding or painting, the robot may hold the workpiece to offer support and stability. The robot’s primary role is to adjust its position to match the operator’s movements, ensuring that the task is completed with the utmost precision and accuracy. This level of collaboration is appropriate for tasks that demand close coordination and communication between robots and humans. It can increase efficiency and productivity by enabling the operator and the robot to work together to complete the task at hand.

### 3.2. Work Roles

In the industrial world, human operators and robots are assigned different roles depending on the task at hand [24]. Completing a task in the HRC system requires the human operator and the robot to completely fulfil their individual and shared responsibilities. Even if they work independently, their tasks are interdependent, and both are crucial to the overall task’s success. The pace of the task can be set individually or mutually agreed upon by both parties. There are two types of relationships between the human operator and the robot: master–slave or peer-based. In a master–slave relationship, the master sets the pace, and the slave follows. In a peer-based relationship, both have equal decision-making power and can collaborate to set the pace. When designing and implementing HRC systems, these relationships are essential to consider as they can impact the system’s effectiveness [25]. According to the classification in [22], The HRC system has three distinct types of role that can be assigned to either human operators or robots for examination.

Supervision: Collaboration between a human operator and a robot at the supervision level is characterized by a master–slave relationship, in which the human operator takes on the role of the master. The human operator is responsible for supervising and directing the actions of the robot, which functions as the slave. This level of collaboration is best suited for tasks that demand a high degree of precision and accuracy, where the human operator must closely monitor the robot’s actions. For example, in a manufacturing process, the human operator may use a joystick or other controls to direct the robot’s movements as it performs tasks like welding or cutting. The robot acts as an extension of the operator’s body, responding to their commands and movements. By enabling the human operator to perform tasks with greater precision and accuracy than they could alone, the supervision level of collaboration can boost efficiency and productivity. However, it can also increase the risk of errors or injuries if the human operator lacks adequate training or experience.Peer: When humans and robots collaborate at the peer level, they have an equal say in decision-making. This type of collaboration is ideal for tasks that require flexibility and adaptability, since both parties must work together to maintain the work rate. For instance, in a warehouse, a human operator may work alongside a robot to transfer items to different locations. The robot may detect obstacles or environmental changes, and the human operators may need to cooperate with the robot to adjust their movements or alter their route. This type of collaboration boosts efficiency and productivity by enabling humans and robots to complete tasks more swiftly and effectively than they could alone.Subordinate: At the subordinate level of collaboration, the interaction between the human operator and the robot is defined as a hierarchical relationship, where the robot acts as the lead. The robot is responsible for guiding and overseeing the actions of the human operator, who acts as the follower. This level of collaboration is appropriate for tasks that require a high level of automation and independence, where the robot has more advanced knowledge or skills than the human operator. For instance, in a medical environment, a robot could be utilized to complete a surgical procedure, while a human operator assists with minor duties like adjusting instruments or supplying materials. The robot would be accountable for directing the overall operation and ensuring that it is conducted safely and efficiently. The subordinate level of collaboration can enhance productivity and minimize the risk of human error by allowing the robot to complete tasks that are challenging or impossible for humans to perform. However, it also raises ethical and safety concerns.

### 3.3. Safety Control Modes

Ensuring the safety of human operators is crucial for the successful implementation of HRC systems. In [26,27], human errors, environmental conditions, and engineering faults have been identified as potential sources of failure in the human–robot working area. To ensure the safety of human operators and robots during work processes, the International Organization for Standardization (ISO) has established safety standards. ISO 10218-1 and ISO 10218-2 are two of these standards, which provide guidelines for the safe installation and operation of robotic systems. These guidelines aim to prevent injury to human operators [28]. Another standard, ISO 15066, was released in 2016, and it provides enhanced safety control modes for the integration of human–robot collaboration (HRC) systems, focusing on factors such as force and speed [29]. To ensure a safe work environment for both human operators and robots, it is highly recommended to implement mandatory safety modes in the HRC system installation. This involves incorporating safety measures, such as emergency stop buttons, protective barriers, and monitoring systems that can instantly detect and respond to any potential hazards. By following these safety standards, the risks associated with HRC systems can be greatly reduced, making it possible for human operators and robots to work collaboratively and safely in industrial settings. The safety tools classification is summarized as follows:Safety-monitored stop: A safety-monitored stop is an essential safety feature in HRC systems that prioritize the safety of human operators working alongside robots. When activated, the robot will come to a halt immediately when a human operator enters a defined safety area. This area can be designated by safety sensors or other monitoring devices installed in the work environment. The safety-monitored stop mode is necessary to prevent accidents and injuries that may occur when human operators are near robots during work processes. By stopping the robot’s movements when a human operator is detected within the specified safety area, the risk of collision or other forms of contact between the two can be minimized. This safety mode is often used in conjunction with other safety measures, such as protective barriers, emergency stop buttons, and safety monitoring systems, to ensure the safe and efficient operation of HRC systems, allowing for productive collaboration between human operators and robots.Hand guiding: In HRC systems, hand guiding is a mode of operation that empowers human operators to manually maneuver the robot without external force. This mode is especially beneficial in situations where the robot requires guidance to execute a specific task or operate within limited space with precision. In the hand-guiding mode, the robot’s movement is controlled directly by the operator using a joystick or pendant device, enabling the operator to make precise adjustments as necessary. Hand guiding is a versatile mode that enables closer collaboration between humans and robots, making it possible to perform tasks that were previously challenging or impossible to automate using traditional programming methods. By enhancing the flexibility and adaptability of HRC systems, the hand-guiding mode makes it possible to use them in an extensive range of industrial applications. Nonetheless, it is important to exercise caution while using this mode to ensure safety.Speed and separation monitoring: The safety mode of operation in HRC systems called speed and separation monitoring is crucial in preventing collisions or accidents between robots and human operators. It achieves this by limiting the robot’s force and speed within designated safety zones to safe levels. Sensors equipped on the robot monitor the distance between it and the human operator, as well as the speed and force of its movements. If the robot gets too close to the human operator, the sensors trigger a safety stop to prevent a collision. Speed and separation monitoring help to prevent accidents and injuries by limiting the robot’s force and speed near the human operator, ensuring a safe working environment for both humans and robots. It is particularly vital in industrial applications, such as assembly lines or collaborative manufacturing processes, where robots and human operators work closely together. Proper safety measures must be put in place to prevent accidents and ensure a safe operating environment.Power and force limiting: HRC systems have a safety mode called power and force limiting that restricts the amount of force and torque exerted by the robot. This mode is programmed to keep the robot within a specific range of force and torque to prevent injury to operators. It is particularly handy when the robot must handle delicate materials or meet humans. By limiting force and torque, this safety mode prevents accidents and material damage. The robot’s maximum power and force in different directions are limited to avoid exceeding these limits, triggering a safety stop to prevent injury or damage. Ultimately, power and force limiting is a vital safety feature for HRC systems as it ensures the robot can work safely alongside human operators without causing harm or damage.

### 3.4. Communication Interfaces

In the field of human–robot collaboration (HRC), the communication and programming techniques used for controlling and operating robots have evolved to become more intuitive and user-friendly. Unlike traditional programming and interfaces which are based on conventional coding methods, the HRC system leverages more intuitive approaches to facilitate communication between humans and robots [30]. To achieve greater efficiency and flexibility in HRC, it is essential to enhance the level of communication between humans and robots. This allows the robot to adapt to all possible human movements and interactions during work, which is crucial for achieving seamless collaboration between humans and robots.

The latest communication techniques, which include body gestures, facial and eye tracking, voice commands, and haptic interfaces [22,31], aim to enable more natural and intuitive interaction between humans and robots. For example, body gestures, such as pointing or waving, can trigger specific actions. With facial and eye tracking, human facial expressions and eye movements are detected and interpreted, providing the robot with contextual cues for responding appropriately. Voice can issue commands to the robot, while haptic interfaces can be used to provide tactile feedback, such as vibrations or pressure, to the human operator.

## 4. Smart Manufacturing

During the era of Industry 3.0, automation was centered around streamlining the production process and monitoring the different components involved, using sensors and actuators. This allowed the human operator to closely observe the manufacturing process and make minor adjustments to the working environment as needed. The goal was to enhance efficiency and accuracy while still relying on human oversight. However, this approach had its limitations, and as the world transitioned to Industry 4.0, more advanced and collaborative approaches were introduced [32]. With the advent of the digital age, industries have begun to explore new approaches that leverage technology to achieve more efficient and effective operations. One such approach is smart manufacturing, which involves the use of advanced technologies such as artificial intelligence and robotics to perform complex tasks with a higher degree of precision and accuracy. Smart manufacturing enables higher quality control, improved safety measures, and greater flexibility in production processes. In contrast to Industry 3.0, where automation solutions focused on automating the production process and relied heavily on human operators to observe and adjust the working environment, Industry 4.0 places greater emphasis on digitalization and technology to optimize manufacturing processes. By embracing this approach, companies can achieve greater efficiency and productivity while minimizing energy consumption and operating costs, resulting in more sustainable and digitally integrated production. Furthermore, with the help of data analytics and customer feedback, manufacturers can tailor their products and services to meet customer preferences, resulting in greater customer satisfaction [33,34,35]. Industry 4.0 enhancements have provided interesting terminologies, outlined in Figure 2.

Currently, manufacturers are focusing on enhancing production levels by integrating cyber and physical systems, smart manufacturing, and predictive maintenance. For instance, industrial applications in Industry 4.0 are using augmented reality for assembly guidance and virtual reality for design and simulation. Therefore, the enhanced company will gain a competitive edge over its rivals and strengthen its position in the market.

Data exchange, value-added services, digitization and customized products and the green industry are the future of the industrial world [36]. These landscapes are enabled in smart manufacturing through four main approaches: Internet of Things (IoT), cloud computing, big data and analytics [37]. As a result, Industry 4.0 adoption can integrate the whole manufacturing system, resulting in smart manufacturing, smart working, and smart products and services [38].

### 4.1. Smart Manufacturing Technologies

Smart manufacturing systems incorporate numerous advanced technologies to improve work processes and environments. Four technologies have been identified as pivotal to the implementation of the Human–robot Collaboration (HRC) system in smart manufacturing. These technologies play a significant role in improving the performance of the HRC system, ultimately leading to enhanced efficiency and productivity in smart manufacturing.

Artificial intelligence (AI) and Collaborative Robots (Cobot), Nowadays, AI is considered a Game-Changer in manufacturing sectors. Both Machine Learning (ML) and Deep Learning (DL) approaches are utilized in the Industry 4.0 age to maximize production in certain and guaranteed ways [39]. By leveraging AI, smart manufacturing has the potential to revolutionize the industrial sector by enabling unprecedented levels of automation, optimization, and flexibility. AI algorithms can process and analyze vast amounts of data generated by sensors, machines, and other sources, extracting valuable insights and patterns that can help manufacturers make better decisions and optimize their operations. Through the integration of manufacturing and information communication technologies, AI-powered smart manufacturing systems can facilitate seamless communication, coordination, and collaboration between different parts of the manufacturing process, improving overall efficiency, quality, and productivity [40]. AI encompasses a range of theories, methodologies, technologies, and practical applications that are aimed at enhancing human intelligence. In addition to the artificial technologies such as ML, DL, reinforcement learning and decision-making, there are initiative applications that are driven by AI and are highly recommended in the industry. For instance, machine vision and recommendation approaches provide insights and hints that may not be otherwise considered by human operators [41].

In the current context of human–robot collaboration (HRC) in smart manufacturing environments, the adoption of AI is facilitating advanced learning processes that enable humans and robots to learn together at a human–human level. In recent years, there has been an increasing need for human operators to collaborate with robots in assembly tasks. While researchers have been developing feature-based approaches to enhance this collaboration, these approaches require a significant amount of manual effort for feature design, and data labelling, and often overlook task contexts. In [42], dual deep-learning input and automated labelling approaches have been introduced to streamline the feature-learning processes and reduce the amount of training effort required. The proposed approach was thoroughly scrutinized by the authors, considering essential factors such as the human–robot interface, feature extraction complexity, task context, and accuracy. Human behaviour data was collected through webcams as part of the human–robot interface. Additionally, the approach required less effort for feature extraction during the modelling phase. Furthermore, using a dual input approach with images, the method produced exceptional accuracy rates for human–robot collaboration scenarios.

Augmented reality (AR): AR is a distinguished and promising technology that can be utilized in smart manufacturing environments. AR is providing support and association information for the human operator which can increase his/her awareness during the work, especially considering assembly tasks and systems design phases [34,43].

In [44], the authors investigated interesting aspects of how AR is utilized with HRC systems such as in safety, guidance, and quality control. The use of augmented reality (AR) is improving safety for workers collaborating with robots. AR can display visual warnings, show the robot’s movements, and help visualize the workspace. These visual warnings could be alerts when the robot is starting or stopping, emergency stop alerts, or other general warnings. The workspace visualization includes two holographic cubical safety volumes, one green and one red, which indicate where it is safe for the human operator to work and where the robot will move. Warning signs and safety zones are also placed on the workpiece and the ground to increase the worker’s awareness of the robot’s actions. AR technology can also assist operators during the assembly process by projecting textual and 3D holographic instructions directly onto the workpiece or a virtual slate. This technology aids in identifying crucial areas, indicating the appropriate placement of parts, and showcasing the required tools and components for a specific task. The system provides real-time updates based on the operator’s actions, identifies completed tasks, and offers comprehensive information and 3D holographic animations. On the other hand, the HRC system that incorporates AR technology can detect defects in parts being handled and alert the operator for inspection. The system projects a blinking marker on the defective part to signal human intervention is necessary. Additionally, the system can verify if the parts are correctly positioned and if all screws have been inserted and inform the user if there is any discrepancy. If an error occurs, a notification is displayed in the same area used for providing instructions. Quality control can also be enhanced in this context.

Digital twin (DT): Digital Twin technology has become an essential instrument in establishing sustainable manufacturing practices within the realm of smart manufacturing. This technology generates a digital replica of the physical shop floor, enabling manufacturers to acquire real-time data on their production process. This, in turn, makes it possible for them to identify opportunities for sustainable improvements, such as reducing waste, optimizing energy consumption, and improving the efficiency of materials. Digital Twin technology is also compatible with other network–physical integration technologies such as virtual and augmented reality, as well as simulation. This integration empowers manufacturers to create advanced simulation models of their production processes and test various sustainable initiatives. Virtual and augmented reality technologies provide real-time data and guidance to workers, enabling them to carry out sustainability-related tasks such as identifying and segregating recyclable materials. By providing manufacturers with real-time data, Digital Twin technology has become a vital component of smart manufacturing, and its integration with other technologies can create more sustainable production environments [45,46].

To create an efficient and cost-effective HRC system for smart manufacturing, it i9s crucial to carefully design and analyse the system. Digital Twin (DT) technology is essential in improving the interaction between humans and robots, allowing real-time monitoring and dynamic decision-making. However, the challenge lies in creating a DT model that accurately represents the collaborative scenario and relationship between physical components. Additionally, maintaining system consistency can be difficult in sustainable smart manufacturing. In response to these challenges, the authors of [47] propose a four-tuple DT model for HRC systems, which includes a human model, robot model, collaborative environment model, and collaboration relation model. The objective of this model is to overcome the obstacles encountered when constructing a proficient HRC system in smart manufacturing. To achieve this goal, the proposed approach concentrates on resolving the challenges associated with optimizing DT, specifically task allocation, path planning, and layout optimization. Additionally, the method considers the consistency of the HRC system while evaluating the model’s operational process. Further scrutiny is necessary to evaluate the accuracy of the entire system in DT models for HRC systems.

### 4.2. HRC in Smart Manufacturing: Industrial Cases

#### 4.2.1. Food Industry

The food industry plays a key role in the European economy, as some shops produce up to 10,000 meals daily [48]. The manufacturing cycle consists of three major stages: farming, production and, finally, ready meals to be sent to the market. Food sector leaders are focusing on transforming the business strategy to be based on demand. This was especially important in the early stages of the COVID-19 pandemic, which negatively impacted supply chain resilience [49]. Therefore, the emergence of digitalization and Industry 4.0 technological contributions are highly required to lead the transformation of food production in order to enhance the sustainability of this sector [50].

Robotics in agriculture is enhancing the collection of information about plants, soil and crop growth. The implementation of sensors is increasing the system’s reliability through intelligent packaging, as sensors are built to provide real-time data about the expiry date of the products [51]. Fruit harvesting is utilized by employing a robot attached to a gripper camera to perform both picking and inspection processes [52]. By integrating an image processing system with the camera, quality control assurance can be enhanced. In addition, installing a vision system on the cobot will enhance consumer confidence that this food is safe and clean, as the camera will be able to detect foreign bodies, such as glasses or plastic. According to [48], in catering facilities, there are several processes (e.g., cooking, baking), and the production challenge lies at the end of the line. Food is processed in this area through manual steps, which are light and can be performed by humans, but they require a high level of repetitive ability, which the human worker lacks at this point.

The food industrial sector requires continuous advancements and developments. Challenges may arise due to the adoption of Industry 4.0 technologies and trusting industrial robots to collaborate with a human operator to perform tasks. However, a delay in implementing these systems will delay the opportunity to benefit from these technological advances, and therefore no tangible change will occur in the industrial sector. HRC system adoption will gradually ensure production processes are working ideally, considering that some tasks cannot be automated and require human expertise, such as feeding machines with components to keep the work continuous.

#### 4.2.2. Automotive Industry

The automotive is the largest industrial sector in the world. Considering the UK only, 3.7 million employees are working in the automotive sector and the economic contribution to the UK economy is about $26 billion [49]. In the automotive industry, assembly cells are playing an important role, where 83% of production units involve assembly tasks [22]. However, some manual operations still need more flexibility and robustness to be performed efficiently; thus, relying on the industrial robot to perform these tasks alone may not be a practical solution as human abilities cannot be fully replaced [53]. Therefore, the focus is to combine the abilities of both humans and robots to work in collaboration, while safety is assured to prevent any accident during the work [29].

From [9], in the assembly stage, the collaborative robot is responsible for the screwing task through sensing integration with a human operator who will be able to share the work area and task. Installing the vision system also allows the collaborative robot to collect information about the working environment and the human intentions that will be used for further improvements such as path planning and human movement predictions. As a result, the implementation of the HRC system is demonstrating the necessary capacity to perform complex tasks.

## 5. Key Findings and Future Research Directions

In the era of Industry 4.0, enhancing manufacturing productivity, efficiency, and cost savings through technologies such as AI, robotics, and IoT is crucial for businesses to remain competitive [54]. However, successfully navigating the digital transformation journey requires more than just implementing new technologies. It demands a fundamental shift in mindset throughout the entire organization, starting with strong leadership commitment and a clear vision for the future [55]. Leaders need to understand the potential of digital technologies and effectively communicate their benefits to the organization. Fostering a culture of innovation, agility, and continuous learning is essential for supporting digital transformation. This involves encouraging experimentation, rewarding risk-taking, and creating an environment where employees feel empowered to embrace new technologies. Engaging employees early on in the transformation process, gathering their feedback, and addressing their concerns are important steps. Comprehensive training programs and upskilling initiatives are necessary to equip employees with the skills needed to work alongside new technologies. Starting with pilot projects allows companies to test new technologies in a controlled environment before scaling up, demonstrating the value and benefits of digital transformation [56]. Collaboration with technology providers, research institutions, and industry peers facilitates knowledge sharing and keeps companies updated with the latest trends and best practices. Effective change management practices, such as developing a clear strategy, communicating the purpose and benefits of the transformation, and establishing regular and transparent communication channels, are critical for success. Emphasizing data-driven decision-making, investing in data analytics capabilities, and continuously evaluating and improving digital initiatives are important factors to consider. By addressing these challenges, companies and industries can successfully navigate the digital transformation journey and leverage the benefits of HRC systems in smart manufacturing within the age of Industry 4.0.

Incorporating AI, Cobots, AR, DT, and HRC in smart manufacturing optimizes data processing, control operations, and production efficiency. These methodologies facilitate data digitization, making it easier to manage, analyze, and utilize. Smart systems provide real-time data insights, leading to informed decisions, reduced waste, cost savings, and increased productivity. Furthermore, smart manufacturing integrates business units, such as supply chain management, customer service, and production, improving collaboration and data management. This contributes to high-quality production, better product traceability, and higher customer satisfaction [57].

Implementing HRC in manufacturing offers a promising alternative to traditional automation systems, reducing complexity and enabling efficient collaboration between humans and robots. User-friendly interfaces facilitate intuitive interactions [58]. HRC systems can enhance efficiency, productivity, and flexibility in production processes, shaping the future of manufacturing [22].

However, the complexity of collaborative robot technology limits its current use to simple production processes, affecting operators’ confidence and decision-making in critical situations. Designing and perceiving human–robot roles is crucial, and safety and accessibility should be maintained [59]. Constructing an integrated HRC system, comprising collaborative robots, human operators, sub-systems, such as vision or sensing systems, and machine learning or deep learning approaches, necessitates a thorough understanding of the management of such complex and advanced working systems. These systems require specialized skills for operation, and some operators may lack the necessary experience or expertise to effectively handle these integrated technologies. The complexity involved can pose challenges for operators who are unfamiliar with these technologies, making training time-consuming and requiring significant investments in education and skill development [56]. Moreover, for companies with high employee turnover, providing regular and comprehensive training programs becomes a bottleneck in ensuring new employees can adapt to these changes. This can be costly and demand ongoing efforts to keep operators updated with evolving technology. Furthermore, it is essential to acknowledge that implementing AI approaches in smart manufacturing environments raises ethical and legal concerns. Safeguarding data privacy and security, addressing algorithmic biases, and complying with regulations related to safety and workers’ rights are critical considerations [60]. Manufacturers must navigate these challenges and ensure that technology implementation aligns with ethical guidelines and legal frameworks [54].

## 6. Conclusions

In this extended version of our work, we have focused on Human–Robot Interaction (HRI) and discussed the concept of Human–Robot Collaboration (HRC) as a complete working system, including its definition, classification, and characterization. We have emphasized the importance of designing the structural components of the HRC system. Furthermore, our paper has explored the integration of HRC with smart manufacturing, with Artificial Intelligence (AI), Collaborative Robots (Cobots), Augmented Reality (AR), and Digital Twin (DT) technologies that have emerged in the era of Industry 4.0. Through examples from the food and automotive industries, we have demonstrated how collaborative robots, equipped with intelligent sensing and vision systems, are implemented to enhance efficiency in these sectors. The successful implementation of collaborative robot systems in these industries highlights the significance of HRC in current manufacturing practices. Collaborative robots hold great promise in leveraging manufacturing efficiency by combining the knowledge and expertise of human operators with the capabilities of robots. However, it is essential to address the current challenges associated with HRC, including complexity, rigidity, safety, and interfacing. Extensive research is required to develop stable and intuitive solutions that can be adapted to various industrial domains [61].

The collaboration between human operators and robots has the potential to revolutionize manufacturing industry, enabling more flexible, efficient, and effective production systems. For instance, integrating in-process quality control into HRC systems represents a promising research direction, as it can improve production processes while maintaining strict product quality standards. By pursuing these research directions, we can advance the understanding and implementation of HRC systems in industrial settings, driving innovation and improving operational efficiencies across diverse manufacturing sectors. Ensuring the safety levels of HRC systems and carefully designing the roles of humans and collaborative robots will be critical for their successful adaptation in manufacturing environments, allowing for consistent production even in the presence of technical issues or sudden changes in the working environment.

## Figures and Tables

**Figure 1 sensors-23-05663-f001:**
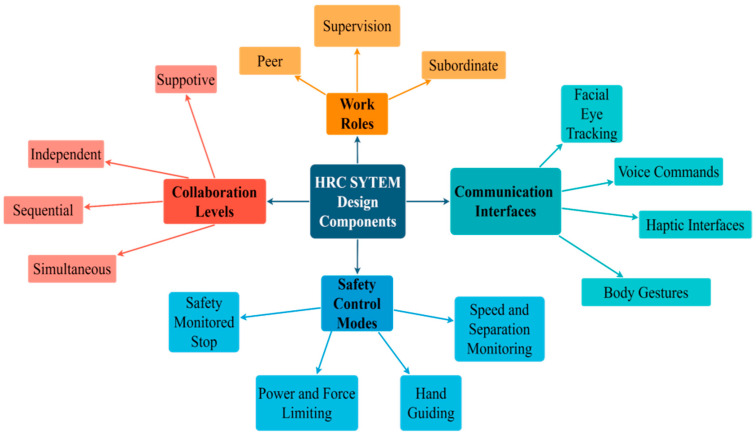
Structural components of HRC system.

**Figure 2 sensors-23-05663-f002:**
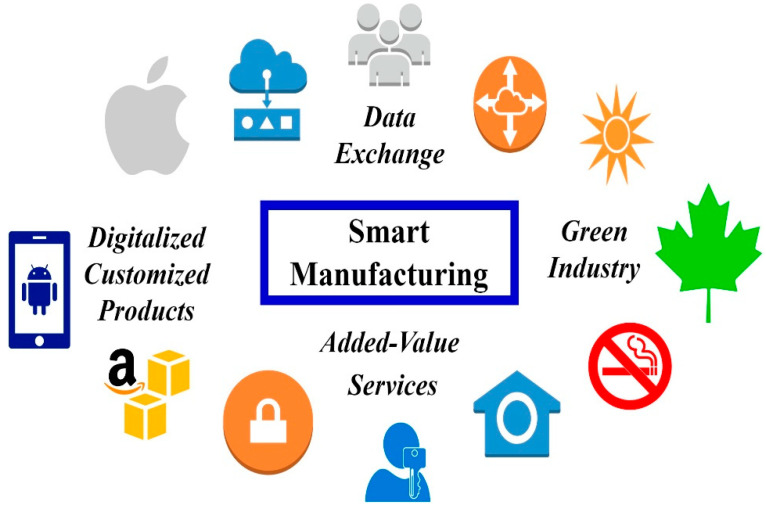
Smart manufacturing technological terminologies.

**Table 1 sensors-23-05663-t001:** Review protocol.

Protocol	Description
Objective	Emphasize the importance of human–robot collaboration systems in smart manufacturing.
Research Questions	Q1: What are the current challenges and limitations associated with integrating the HRC system in smart manufacturing? Q2: How can collaborative systems between humans and robots be effectively implemented in smart manufacturing? Q3: How does the implementation of key technologies with human–robot Collaboration systems affect manufacturing flexibility, efficiency, and sustainability?
Databases	SCOPUS, IEEE Explore, SPRINGER link
Relevant Literature	Focused on papers related to smart manufacturing, production process enhancement, and human–robot interactions in the industry.
Inclusion Criteria	Publications from the last 5 years, English language, literature reviews with industrial applications. “Human–robot Collaboration” OR” Collaborative Robots” AND” Smart Manufacturing” OR “Industrial applications” OR “Industry 4.0”
Keywords	Literature review, smart manufacturing, human–robot collaboration, human–robot collaboration applications in smart manufacturing, industry 4.0, AI
Specific Journals	Mechatronics, CIRP Annals—Manufacturing Technology, Robotics and Autonomous Systems
Software/Tools	Microsoft Word (for writing), Endnote (for reference management)
Data Analysis	Qualitative analysis
Limitations	Identified limitations of human–robot collaboration systems in smart manufacturing and suggested future research directions
Citation style	MDPI referencing style

**Table 2 sensors-23-05663-t002:** HRI interaction on shared content.

	Interaction			
Shared Content	Coexistence	Synchronization	Cooperation	Collaboration
**Work Task**		**×**		**×**
**Direct Contact**		**×**		**×**
**Simultaneous process**	**×**		**×**	**×**
**Workspace**		**×**	**×**	**×**
**Sequential process**		**×**	**×**	

## Data Availability

Not applicable.

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
