# Peer review of "Human–Robot Collaborations in Smart Manufacturing Environments: Review and Outlook†"

_sensors, 2023, doi:10.3390/s23125663_

Round 1

Reviewer 1 Report

While the manuscript provides a good review on outlook, I have the following concerns, most notably a lack of critical reasoning with certain aspects that I believe are critical to the implementation and adaptation of human-robot collaboration:

1. I would like more discussion on the challenges of adaptation of certain technology. E.g., discussion on operator training requirements in a smart manufacturing environment. Integrating technology to aid knowledge transfer could be a key, especially in manufacturing industries with high employee turnover. Honeywell, as an example, is a company that provides virtual reality and augmented reality training products to enhance operator and engineer understanding, which also allows improved operator training for incidents that may not occur as often (emergency scenarios and the like).

2. I would like a discussion on the challenge of digital transformation, i.e., mindset change of the company and what can be done. Human-robot collaboration is important, however, to achieve true success in deployment and implement, the entire company / industry needs to be on board. What are successful strategies that companies and/or industries have used in the past?

3. The authors do not provide an indication of what they see as the most promising direction for future work, as well as the most pressing challenges. It is important to present viewpoints of the author’s themselves, so that their critical reasoning as experts in the field can be judged and reviewed. However, this is lacking in the conclusion.

Note: In order to address these comments, the authors may need to make other parts of the manuscript more concise, as the manuscript appears lengthy.

Author Response

1st Reviewer Response

First of all, we truly appreciate your kind efforts and the time you spent reviewing my manuscript.

we would like to reiterate that the article is an extended version of my conference paper that we presented at ICAC2022 in September 2022 in Bristol, UK. In the conference paper, we aimed to highlight that the implementation of Human-Robot Collaboration systems is an interesting research area that has the potential to revolutionize the way manufacturers perceive robotics systems in smart manufacturing.

The extended version continues to focus on the same topic as the conference paper. However, we delve deeper into the technologies employed in smart manufacturing that are necessary to make the implementation of Human-Robot Collaboration systems more feasible and practical. Therefore, we are delighted to provide you with a response for each point you mentioned, which you can find below:

Q1:  I would like more discussion on the challenges of adaptation to certain technology. E.g., a discussion on operator training requirements in a smart manufacturing environment. Integrating technology to aid knowledge transfer could be a key, especially in manufacturing industries with high employee turnover. Honeywell, as an example, is a company that provides virtual reality and augmented reality training products to enhance operator and engineer understanding, which also allows improved operator training for incidents that may not occur as often (emergency scenarios and the like).

Answer:

 we concur with your viewpoint as we overlooked mentioning the existing limitations that arise during the implementation of Human-Robot Collaboration (HRC) systems in Smart Manufacturing, incorporating key integrated technologies. In the previous discussion, we focused on how HRC systems can enhance work flexibility, simplify the interaction between humans and robots, and alleviate functionality issues found in conventional robotic systems. However, it is crucial to address the current limitations of HRC implementations, especially when integrating trending technologies to boost productivity and create more advanced working systems.  Please Check [Section 5, page 14,15]

Q2: I would like a discussion on the challenge of digital transformation, i.e., mindset change of the company and what can be done. Human-robot collaboration is important, however, to achieve true success in deployment and implement, the entire company / industry needs to be on board. What are successful strategies that companies and/or industries have used in the past?

Answer:

we totally agree with your suggestion and question about the digital transformation in the industry as the gap between knowledge and the real industrial world is still large. Therefore, the research should discuss this side effect of the technologies implementation especially with HRC systems. Please Check [Section 5, page 14,15]

Q3. The authors do not provide an indication of what they see as the most promising direction for future work, as well as the most pressing challenges. It is important to present viewpoints of the author’s themselves, so that their critical reasoning as experts in the field can be judged and reviewed. However, this is lacking in the conclusion.

Answer:

In the discussion and key findings of this research, we have highlighted several future work directions that hold promise in this field. One notable direction is the exploration of industrial applications where HRC systems are implemented. This area presents significant potential for further research and offers ample opportunities to expand the application capacity of these systems. Moreover, there are additional promising avenues for future work that warrant attention. Please Check [Section 6 page 15]

Reviewer 2 Report

I think that the authors have to give a more comprehensive review and outlook of Human-Robot Collaborations in Smart Manufacturing Environments.

Author Response

2nd Reviewer Response

First of all, we truly appreciate your kind efforts and the time you spent reviewing my manuscript.

we would like to reiterate that the article is an extended version of my conference paper that we presented at ICAC2022 in September 2022 in Bristol, UK. In the conference paper, we aimed to highlight that the implementation of Human-Robot Collaboration systems is an interesting research area that has the potential to revolutionize the way manufacturers perceive robotics systems in smart manufacturing.

The extended version continues to focus on the same topic as the conference paper. However, we delve deeper into the technologies employed in smart manufacturing that are necessary to make the implementation of Human-Robot Collaboration systems more feasible and practical. Therefore, we are delighted to provide you with a response to your comment, which you can find below:

Comment:

I think that the authors have to give a more comprehensive review and outlook of Human-Robot Collaborations in Smart Manufacturing Environments.

Answer:

In the article, the workflow was chosen to cover the Human-Robot Interaction levels with more focus on the collaboration interaction between the human and robot as one of the research interests. After that, we considered the Human-Robot Collaboration system design and the aspects that should be considered when it comes to the system’s design phase. Considering the smart manufacturing area, we covered key technologies where the HRC systems are utilized with these technologies and examples were provided. the food and autorotative industries are two case studies that were addressed to point out how collaborative robots are employed nowadays in the two biggest industrial sectors.

In sections 5 and 6, the modified material we added intends to cover the digital transformation challenges in smart manufacturing and the challenges of implementing HRC in industry. [Please check sections 5 and 6 pages 14 and 15 of the paper]

Reviewer 3 Report

The article is very interesting and before accepting this article, please address the following questions and consider my suggestions as listed below.

1.     Would you please add a section to introduce the methodology used in the article?

2.     What are the industries you observed, since you state the work presented here focuses on the design of HRC systems, with particular attention given to the various levels of Human-Robot Interaction (HRI) in the Abstract? Are they Section 4.2.1(Food Industry)? and Section 4.2.2(Automotive Industry) only? In case of them, you need to state in the Abstract.

3.     In section 6(Conclusion), what four major technologies emerged in the age of Industry 4.0 since you highlighted smart manufacturing with a focus on four major technologies?

Author Response

3rd Reviewer Response

First of all, we truly appreciate your kind efforts and the time you spent reviewing my manuscript.

we would like to reiterate that the article is an extended version of my conference paper that we presented at ICAC2022 in September 2022 in Bristol, UK. In the conference paper, we aimed to highlight that the implementation of Human-Robot Collaboration systems is an interesting research area that has the potential to revolutionize the way manufacturers perceive robotics systems in smart manufacturing.

The extended version continues to focus on the same topic as the conference paper. However, we delve deeper into the technologies employed in smart manufacturing that are necessary to make the implementation of Human-Robot Collaboration systems more feasible and practical. Therefore, we are delighted to provide you with a response for each point you mentioned, which you can find below:

Q1: Would you please add a section to introduce the methodology used in the article?

Answer:

we will definitely include a methodology section to outline the approach we followed in conducting this research article. Please check Section 1.1, pages 2 and 3 added to the paper.

Q2: What are the industries you observed, since you state the work presented here focuses on the design of HRC systems, with particular attention given to the various levels of Human-Robot Interaction (HRI) in the Abstract? Are they Section 4.2.1(Food Industry)? and Section 4.2.2(Automotive Industry) only? In case of them, you need to state in the Abstract.

Answer:

In this article, the focus is on discussing industrial fields where Collaborative Robots (Cobots) are employed to perform specific applications as they are programmed to do. The automotive and food industries are considered the two largest sectors in the industrial world where the implementation of HRC systems can have significant opportunities for utilization in production and impact the entire manufacturing environment. The food and automotive industries are stated now in the abstract. Please check the abstract on page 1

Q3:   In section 6(Conclusion), what four major technologies emerged in the age of Industry 4.0 since you highlighted smart manufacturing with a focus on four major technologies?

Answer:

The four key technologies utilized in smart manufacturing, as mentioned in the research, are Artificial Intelligence (AI), Collaborative Robots (Cobots), Augmented Reality (AR), and Digital Twin (DT). we will make sure to mention these technologies again in the Conclusion section of the article.  Please check Section 6 page 15

Round 2

Reviewer 2 Report

Paper can be accepted in present form